# Improving the Modeling of Extracellular Ligand Binding Pockets in RosettaGPCR for Conformational Selection

**DOI:** 10.3390/ijms24097788

**Published:** 2023-04-24

**Authors:** Fabian Liessmann, Georg Künze, Jens Meiler

**Affiliations:** 1Institute for Drug Discovery, Medical Faculty, Leipzig University, 04103 Leipzig, Germany; fabian.liessmann@medizin.uni-leipzig.de (F.L.);; 2Department of Chemistry, Vanderbilt University, Nashville, TN 37235, USA; 3Center for Structural Biology, Vanderbilt University, Nashville, TN 37235, USA; 4Center for Scalable Data Analytics and Artificial Intelligence, Leipzig University, 04105 Leipzig, Germany

**Keywords:** homology modeling, Rosetta, RosettaGPCR, pocket refinement, ligand docking, drug discovery

## Abstract

G protein-coupled receptors (GPCRs) are the largest class of drug targets and undergo substantial conformational changes in response to ligand binding. Despite recent progress in GPCR structure determination, static snapshots fail to reflect the conformational space of putative binding pocket geometries to which small molecule ligands can bind. In comparative modeling of GPCRs in the absence of a ligand, often a shrinking of the orthosteric binding pocket is observed. However, the exact prediction of the flexible orthosteric binding site is crucial for adequate structure-based drug discovery. In order to improve ligand docking and guide virtual screening experiments in computer-aided drug discovery, we developed RosettaGPCRPocketSize. The algorithm creates a conformational ensemble of biophysically realistic conformations of the GPCR binding pocket between the TM bundle, which is consistent with a knowledge base of expected pocket geometries. Specifically, tetrahedral volume restraints are defined based on information about critical residues in the orthosteric binding site and their experimentally observed range of C_α_-C_α_-distances. The output of RosettaGPCRPocketSize is an ensemble of binding pocket geometries that are filtered by energy to ensure biophysically probable arrangements, which can be used for docking simulations. In a benchmark set, pocket shrinkage observed in the default RosettaGPCR was reduced by up to 80% and the binding pocket volume range and geometric diversity were increased. Compared to models from four different GPCR homology model databases (RosettaGPCR, GPCR-Tasser, GPCR-SSFE, and GPCRdb), the here-created models showed more accurate volumes of the orthosteric pocket when evaluated with respect to the crystallographic reference structure. Furthermore, RosettaGPCRPocketSize was able to generate an improved realistic pocket distribution. However, while being superior to other homology models, the accuracy of generated model pockets was comparable to AlphaFold2 models. Furthermore, in a docking benchmark using small-molecule ligands with a higher molecular weight between 400 and 700 Da, a higher success rate in creating native-like binding poses was observed. In summary, RosettaGPCRPocketSize can generate GPCR models with realistic orthosteric pocket volumes, which are useful for structure-based drug discovery applications.

## 1. Introduction

With about 800 members in the human genome, G protein-coupled receptors (GPCRs) comprise the largest family of transmembrane proteins [1]. They are expressed in every organ system and are crucial for a plethora of diverse biological processes including visual and olfactory perception, neurotransmission, regulation of blood pressure, hormone secretion, and many more [2,3]. Furthermore, these receptors can sense a wide range of stimuli including light, ions, small molecules, lipids, and even peptides and other proteins [4,5,6]. Corresponding to their wide range of functions and their location at the interface between a cell and its environment, GPCRs constitute a key target for drug discovery and design. With nearly one-third of all FDA-approved drugs targeting the family of GPCRs, they are undoubtedly an important scientific objective [2,7,8]. GPCRs can be divided into six major classes according to their amino acid sequence (class A-F) [9,10]. However, only four classes are found in the human genome (A—rhodopsin-like, B—adhesion and secretin, C—glutamate, and F—frizzled) [11,12]. The biggest group, class A, contains 719 receptors in humans and roughly half of them are sensory (olfactory) receptors [13].

Since the structure determination of bovine rhodopsin [14] and the first human GPCR [15], the number of experimentally determined unique structures has been steadily increasing. To date, the structures of more than 150 GPCRs have been determined and it can be expected that this number will continue to increase. All GPCRs have a common architecture consisting of seven transmembrane helices (7TMs, numbered TM1–7, accordingly), connected by three intra- and three extracellular loops (ICLs and ECLs, respectively) as well as an eighth intracellular helix (H8) [16,17]. The sizes of the extra- and intracellular domains differ from only a few residues up to thousands. Some of the largest extracellular domains are seen for members of the adhesion GPCRs [18,19]. A pocket in the TM bundle, open to the extracellular space, usually represents the endogenous ligand binding site in class A GPCRs. In this paper, the binding site in the middle of the TM bundle is simply referred to as a pocket. The binding of an extracellular ligand to this pocket is the starting point for receptor activation. Upon ligand binding, conformational changes in key residues occur, which leads to an outward movement of TM5 and TM6 on the intracellular side allowing for interaction with G proteins. The eponymous G protein can bind in the emerged intracellular crevice and transmits the signal to downstream effectors. Additional pathways including β-arrestin binding and desensitization of GPCRs have been investigated [1,20,21,22,23]. The topic of biased signaling and explicitly addressing specific pathways is of utmost interest in recent years. Furthermore, the concept of allosteric microprocessors including a multitude of structural conformations changed our understanding of GPCR signaling [24,25,26].

To further investigate the process of signaling and to develop probe and lead molecules for drug discovery, high-quality structural models are needed. For human GPCRs deemed druggable, less than 30% of structures have been determined experimentally, representing a prodigious knowledge gap. While this number is expected to further increase, experimental structures provide static snapshots of very flexible proteins with alternative conformations that might be stabilized with a small molecule. For the remaining GPCRs, modeling approaches including SWISS-MODEL [27] and the Rosetta software suite [28,29] can provide homology models of GPCRs. Furthermore, deep learning methods such as RoseTTA fold [30] or AlphaFold2 [31] have revolutionized structure prediction in recent times. RosettaCM is a Rosetta homology modeling protocol that has also shown good performance for difficult cases when no high-sequence identity template is present. By combining multiple template structures instead of using only one structure, RosettaCM can mitigate the problem of low sequence identity and create high-quality models [32]. RosettaCM has been successfully applied in studying the interaction of various GPCRs with peptides and ligands, respectively [33,34,35,36]. The availability of high-resolution structures and accurate models of GPCRs facilitates enormously the discovery of novel ligands and the investigation of their interactions for structure-based drug discovery [37,38].

An optimized RosettaCM homology modeling pipeline for GPCRs was developed by Bender et al. [39]. This method, called RosettaGPCR, improves the comparative modeling of GPCRs with low sequence identity and can deliver high-quality predictions for the extracellular loop regions. For the modeling process, Rosetta utilizes an energy function, describing the physical and chemical properties of the protein and calculating the energy of all atomic interactions [40]. The solvation energy of membrane proteins is calculated using an implicit membrane solvation model in Rosetta, which, however, comes with some caveats: transmembrane pores and cavities in membrane proteins reside in a continuous hydrophobic membrane slab devoid of water molecules. Consequently, structure optimization tends to decrease the pocket volume to minimize the energy for this part of the protein, which leads to a shrinkage of the binding pocket. This will impair downstream applications such as ligand docking. In addition, the pronounced flexibility of GPCRs suggests that a range of binding pocket geometries and sizes needs to be considered when testing different drugs. As a prerequisite for structure-based computer-aided drug discovery, an accurate receptor model is needed for ligand docking and accurate binding free energy calculations [13,41]. By decreasing the volume of the binding pocket, unrealistic interactions are forced, and larger molecules are discriminated against. As no model is without structural caveat and bias, the prediction of the holo conformation of the receptor binding pocket is a non-trivial task and several approaches have been suggested [42,43].

In this study, we focus on improving the RosettaGPCR modeling pipeline and the subsequent ligand docking applications. The developed new method, RosettaGPCRPocketSize, creates an ensemble of possible pocket geometries and sizes based on a knowledge base of experimental class A GPCR structures. We define the binding pocket by inwards pointing helical residues that frequently interact with ligand molecules in the template structures. Homologous residue positions are identified by using the Ballesteros Weinstein (BW) numbering scheme [44]. In the BW numbering scheme, two numbers separated by a dot indicate the helix and the residue position. The GPCRdb numbering corrects helix bulges and alignment gaps by adding a separator x and a third number. Therefore, the exact position of a residue in the helix can be determined [45,46]. The here-described method, RosettaGPCRPocketSize, utilizes tetrahedral distance ranges to restrict the volume and size of the pocket and prevent pocket shrinkage. A filtering process ensures that generated pocket sizes are diverse for use in ligand docking applications. The restraints can be used in the process of homology modeling to build an ensemble of conformations or to estimate the flexibility of GPCR structures.

## 2. Results

### 2.1. Investigation of Pocket Volume by Distance Measurements between Pocket Lining Residues in Experimentally Determined GPCR Structures

In order to optimize the orthosteric binding pocket and its volume, it is important to understand its geometry, size, and residue composition. Therefore, we evaluated all available experimentally determined GPCR structures in the GPCRdb (as of November 2020). A focus was put on class A GPCRs, which comprise nearly 90% of all GPCRs [11] and more than 80% of all determined structures are from this class. In November 2020, the GPCRdb contained 401 total and 70 unique class A GPCR structures. As some receptors are overrepresented with multiple occurrences in the dataset (such as rhodopsin (OPSD) or the β1-adrenoceptor (ADRB1)), only the structure with the highest resolution was considered. Furthermore, a structure can be in an active, inactive, or intermediate state. Therefore, only inactive state structures were selected because the largest fraction of experimentally determined GPCR structures is in this state.

In this investigation, we selected 34 residues and their respective pairwise C_α_-C_α_ distances to describe the pocket geometry. As the sidechains of residues are flexible and the amino acids vary between different receptors, the C_α_-atom of the amino acids was selected as geometric coordinates. As a side effect, calculating the backbone spanning volume is computationally faster than computing the volume defined by the molecular surface. A previously published analysis of the OPSD structure identified 30 critical residues that line the orthosteric binding site. The selected residues were in a 10 Å sphere around the OPSD ligand retinal and had a surface that was at least 25% accessible to ligands [47]. Later, with more GPCR structures available, the set of critical residues was updated and extended to 44 residues [48]. Based on the published results, we evaluated the residue selection for the current GPCR structure set. All residues that are in a 3.5 Å radius of any pocket-bound ligand atom in the determined structures were selected and their respective BW numbering was compared to the published critical residues. Finally, the most frequently observed residues facing inwards to the pocket were selected, resulting in the 34 selected residues. These residues are mapped onto the snakeplot for rhodopsin shown in Figure 1A.

The C_α_-C_α_ distances between all pairs of selected residues were calculated, yielding 34 × 33/2 = 561 unique distances. For every distance, the median, average, and standard deviations were calculated. Often, the distances between residues on different helices do not vary more than 2 Å, and between related receptors, the differences are very small. In Appendix A all investigated distances are shown.

Instead of calculating the volume considering all 34 selected residues and their flexible sidechains, only the C_α_ atoms of ten residues, which best captured the volume of the pocket, were used as surrogates. Here, we utilized the following residue list: 2.57 × 56, 2.65 × 64, 3.28 × 28, 3.37 × 37, 5.38 × 39, 5.47 × 47, 6.51 × 51, 6.58 × 58, 7.32 × 31, and 7.43 × 42. The volume was calculated using Delaunay triangulation (see Method section). This was done because the residues laying on the same helix are almost collinear and contribute little information to the volume calculation. Therefore, we decided to use the residues with the highest and lowest number on each selected helix to calculate the volume. As TM1 and TM4 are not surrounding the pocket with more than two residues, they were left out. The volume of all previously selected structures was calculated in this way. The volume ranged from 2100 to over 3400 Å^3^, whereas 68% of the receptors have a pocket volume of 2600 ± 200 Å^3^ and 85% are in the range of 2600 ± 300 Å^3^ (see Appendix A).

### 2.2. Experimentally Determined GPCR Structures Sample a Range of Possible Ligand Binding Pocket Sizes, but in Homology Models Pocket Shrinkage Is Observed

For developing RosettaGPCRPocketSize and making sure that biophysically realistic geometries are sampled, we first investigated the size of the orthosteric pocket in different GPCR structures. The pocket volume observed for the inactive state structures of adenosine receptor A2 (AA2AR), ADRB1, β2-adrenoceptor (ADRB2), OPSD, and orexin 1 receptor (OX1R) varies by around ±100 Å^3^ up to ±150 Å^3^ from the median volume, as shown in Figure 2A. For these receptors, the largest number of structures is available as a basis for comparison. It can be seen that these structures explore a range of pocket sizes even for the same specific conformational state. Thus, we conclude that single GPCR models will likely not portray the natural flexibility of the orthosteric binding pocket.

Additionally, when comparing the structures of GPCRs for which both active and inactive state structures were determined, a significant volume change is evident between active and inactive state structures. As shown in Figure 2A, this volume change can be observed for several receptors. For AA2AR, the acetylcholine (muscarinic) receptor 2 (ACM2) and ADRB2, the volume decreases in the active receptor by 153 Å^3^, 181 Å^3^, and 92 Å^3^, respectively, which corresponds to a 4–7% change compared to the pocket volume in the inactive state. For the angiotensin receptors 1 (AGTR1), the decrease is even more pronounced with 496 Å^3^ (21% reduction in total pocket volume in active state). The cannabinoid receptors 1 and 2 (CNR1 and CNR2, respectively) display a wider range of volumes in the active and inactive state, and the pocket volume of CNR2 is even increased in the active state. Noteworthy, the ligands in the CNR2 structures are bulkier compared to the ligands of other GPCR structures investigated, which could explain the larger binding pocket of CNR2. However, we note that only two structures for each conformational state of CNR2 were determined and that one structure (PDB: 6KPC) has a significantly smaller binding pocket. Therefore, more structures for CNR2 would be needed to derive general conclusions about the nature of its ligand-binding pocket [49,50,51]. A small volume increase is also observed for the active state structure of OPSD. However, OPSD is a special case because its ligand retinal is covalently bound and the receptor has a closed binding pocket, resembling more a binding hole. Tight interactions between OPSD and retinal are crucial for the activation mechanism upon photon receiving. This differs from the non-covalent ligand binding modes of other GPCRs. We note that the active OPSD structures in the absence of ligand (e.g., 5W0P [52] with a pocket volume of 2313 Å^3^) were excluded from the analysis. On the other hand, the inactive structures without ligands were included. Normally, a stabilizing ligand is needed for experimental determination of the active conformation. However, in these cases, the determined active OPSD structures without ligands are the result of stabilizing mutations. Summarized, our observations demonstrate a variety of receptor-binding pocket geometries and receptor–ligand interactions exist for GPCRs.

Furthermore, the pocket size depends on whether the receptor was determined in the presence or absence of ligands. If no ligand is present, the binding site is usually larger than if a ligand is present. Examples of this effect are shown in Table 1. Here, the volume and volume changes for the highest quality inactive structure of the muscarinic acetylcholine receptor 4 (ACM4, PDB: 6KP6 and 5DSG) [53,54], ADRB1 (PDB: 4GPO and 4BVN) [55,56], and endothelin receptor type B (ENDRB, PDB: 5XPR, 6IGK) [57,58] with and without ligand are summarized. A shrinkage of around 361 Å^3^ for ACM4, 288 Å^3^ for ADRB1, and 305 Å^3^ for ENDRB, respectively, could be observed as an effect of ligand binding. By contrast, for OPSD (PDB: 2I37 and 1U19), the differences are marginal, and even a slight increase in the pocket volume upon ligand binding is observed [59,60]. As noted, the activation mechanism and binding site nature of OPSD is special and different from that of other class A GPCRs. In general, for capturing the structure of an inactive GPCR, a ligand is needed to stabilize the conformation, ultimately decreasing the binding pocket compared to the apo-state.

In summary, for the GPCRs investigated, a volume change between active and inactive states from as low as 92 Å^3^ up to 496 Å^3^ could be observed. Ligands bound to the receptor change the size and geometry of the pocket accordingly and stabilize a specific conformation. Upon binding a ligand or switching into the active state, the binding pocket usually becomes smaller reflecting increased interactions between receptor and ligand. Even for the same conformational state, structures can show a range of pocket volumes. This effect was recognized and comprehensively investigated for ADRB1 [61]. In conclusion, a receptor does not have one single pocket representation but distinct pocket geometries, even for the same activation state. Thus, we decided that a pocket-size change of ±200 Å^3^ between the different model structures generated in homology modeling is a reasonable representation of the expected volume change dynamics for a receptor.

To evaluate the performance of the RosettaGPCRPocketSize method, we first analyzed the extent of pocket change in the unmodified RosettaGPCR protocol. Receptors from different class A GPCR subfamilies were selected, including adenosine A1 receptor (AA1R, PDB: 5UEN) [62], AA2AR (PDB: 5NM4) [63], ADRB1 (PDB: 4BVN) [56], ADRB2 (PDB: 2RH1) [16], C-C chemokine receptor type 7 (CCR7, PDB: 6XZH) [64], CXC chemokine receptor 2 (CXCR2, PDB: 6LFL) [65], CXC chemokine receptor 4 (CXCR4, PDB: 3ODU) [66], dopamine 3 receptor (D3R, PDB: 3PBL) [67], histamine 1 receptor (H1R, PDB: 3RZE) [68], µ-opioid receptor (MOR, PDB: 4DKL) [69], neurokinin 1 receptor (NK1R, PDB: 6HLP) [70], and sphingosine 1-phosphate receptor (S1P1R, PDB: 3V2Y) [71]. The difference in the pocket volume between the homology model generated by the RosettaGPCR protocol and the experimental structure of a particular GPCR is illustrated in Figure 2B. As can be seen, the pocket volume is decreased in the homology model for all GPCRs investigated and the change can be as high as 640 Å^3^. To further investigate what can be the cause of that pocket shrinkage, we followed the volume change for every GPCR when the experimentally determined structure was iteratively relaxed in the Rosetta energy function. The pocket volume was found to shrink clearly in the first two to three relax iterations after which the pocket volume stayed fairly constant (Figure 2C). Thus, we conclude that the pocket shrinkage seen in the RosettaGPCR models is likely a consequence of the relaxation step inside RosettaCM. To overcome this problem, we constructed special volume-based restraint sets based on known distances and volume ranges as described in the next section.

### 2.3. Design of the Pocket Restraint Set

To control the size of the receptor binding pocket in homology modeling with Rosetta, we used distance restraints. A detailed explanation of the restraint selection procedure is given in the Method section. Out of the 34 residues on the inside of the binding pocket and the resulting 561 residue pairs, 478 pairs connect different helices and were considered as possible distance restraints. To select the best restraint set, several benchmarks with different kinds of restraint geometries and parameters were carried out. For that purpose, the aforementioned twelve GPCR structures were used and the amount of pocket shrinkage in RosettaGPCR was investigated for different restraint sets. In the end, one restraint set of two tetrahedrons with optimized parameter ranges was found most effective in preventing pocket shrinkage. The optimized homology modeling protocol is called RosettaGPCRPocketSize.

First, distance restraint sets forming different kinds of geometries and approximating the pocket volume were tested. These geometries included linear distances, triangles, tetrahedrons, and cubes. The cubic restraints were split further into several sets of restraints. Tested geometries can be found in Appendix A. For the triangular and tetrahedral restraint sets, the triangles spanning the largest surfaces between helices were selected. Based on Heron’s formula, the area of triangles and the volume of tetrahedrons correlates with the length of the edges.

Simple two-dimensional linear distance restraints failed to prevent pocket shrinkage. The impact of single distances was too small to change the volume as a whole. Increasing the restraint weight, however, some specific, but unconnected C_α_ distances were changed, which resulted in the distortion of the pocket structure. On contrary, a restraint set defining a cube has too many distances, which also resulted in a distortion of the pocket geometry. Moreover, a cube is an unfavorable geometric shape in the case of an asymmetric pocket. Additionally, an exhaustive restraint set including all 561 distances between all 34 selected residues was tested. A highly over-fitted pocket, which was similar for all benchmarked receptors and failed to show the receptor-specific differences in pocket shape and geometry, was constructed in this case. We identified restraints connected in the form of a tetrahedral geometry as the overall best choice. This tetrahedron restraint is set up by four Cartesian points connected by six edges.

The tetrahedral restraint set was benchmarked for different selected tetrahedrons and different set sizes—one, two, and four tetrahedrons. The best results were achieved with two tetrahedrons spanning across the whole pocket. Finally, the following residues were selected: tetrahedron 1: 2.60 × 59, 3.40 × 40, 5.38 × 39, and 7.32 × 31; and tetrahedron 2: 2.65 × 64, 4.57 × 57, 5.46 × 461, and 6.48 × 48. The distances are listed in Table 2. An example can be seen in Figure 1B for one geometric tetrahedron. These results show again a so-called Goldilocks effect: Several restraints are better than just a few. However, too many restraints restrict the pocket too tightly; i.e., the goal of achieving different pocket geometries and sizes is not met (see Appendix A).

After benchmarking several parameters for nested restraints in the Rosetta calculation, a parameter range was defined (see Appendix A). The weight of the restraints was set to 2–4, and the reference distance of the restraint was increased from the selected C_α_ atom distance by 3–7% because it was found that this better-prevented pocket shrinkage. A harmonic function was used as the restraint potential with a steepness between 0.4 and 0.8 and a constant y-offset of −1 to −10. These parameter ranges were found optimal in reproducing the size of the reference pockets and allowing some variability of the pocket size in the range of the pocket size deviation found for the GPCR structure set (±200 Å^3^). A figure explaining the effects of selected parameters is given in Figure 3A. The parameters are selected randomly from the defined range in the restraint generation. The six distances defining one tetrahedron have the same parameters, but the two tetrahedrons belonging to one restraint set can have different parameter values. A final volume-filtering step was added to the end of the RosettaGPCRPocketSize protocol. Here, the pocket volume of all generated models is calculated, and the median volume is determined for each separate receptor individually. Models within ±200 Å^3^ to the median volume pass the filter while models with pocket sizes outside of that range are filtered out. This avoids models with extreme pocket size differences in favor of a more realistic pocket size distribution. However, we note that a manual inspection of the final models and their binding pockets can further improve the results.

The best five models according to total energy and pocket volume were selected. To this end, we benchmarked two Rosetta membrane protein scoring functions, membrane_highres_Menv_smooth [32] and franklin2019 [72], in the final model scoring and selection step, but no significant differences in performance were observed (see Appendix A). Using the restraint score in the final selection step, in addition to its usage in the other protocol steps, did not further improve model quality in terms of the amount of pocket opening and was therefore not done in the final RosettaGPCRPocketSize protocol. In the final protocol, the standard membrane energy function without the tetrahedron restraints is used in the last scoring step.

To validate the performance of the new RosettaGPCRPocketSize protocol, we constructed homology models for the receptors in the aforementioned benchmark set and measured the change in pocket volume. The volume shrinkage of the twelve receptors investigated without restraints was in the range of 16 to 637 Å^3^. Two receptors, CXCR2 and NK1R, showed the largest shrinkage with over 600 Å^3^. In contrast, two other receptors, ADRB1 and ADRB2, had the lowest pocket shrinkage, which varied by ±200 Å^3^. Except for CCR7, CXCR2, and NK1R, the determined structure was part of the template set in comparative modeling, but a pocket shrinkage was still observed. By applying the tetrahedral restraints, the shrinkages were decreased to a maximal 473 Å^3^ (see Figure 3C). For several receptors, more precisely AA2AR, ADRB2, and CCR7, an increased pocket volume was observed. However, 3/5 models of AA2AR and 4/5 models of ADRB2, respectively, fulfilled the selected ±200 Å^3^ bandwidth. For CCR7, the volume was increased above the cutoff. For nine out of the twelve receptors, the pocket volume was in the range of ±200 Å^3^ relative to the experimental reference structure. In CXCR2, the pocket shrinkage was reduced from over −600 Å^3^ to only −330 Å^3^ on average. CXCR2 is a member of the chemokine receptors and the natural ligands of this receptor are proteins. Therefore, the orthosteric binding pocket is larger with 3071 Å^3^ in comparison to the 2600 ± 300 Å^3^ average size of inactive class A GPCRs, and a pocket shrinkage is not unexpected. However, for CXCR4, also a member of the chemokine receptor subfamily, the pocket shrinkage could be almost completely prevented. The deviation of the volume improved from −515 to −91 Å^3^ on average.

For ADRB2, only a slightly decreased pocket volume was obtained with the original protocol without restraints. When using RosettaGPCRPocketSize, a slight volume increase of the pocket was observed, which was in the range of the naturally observed variance for this GPCR. This highlights the advantage of RosettaGPCRPocketSize for generating other possible pocket ensembles. When the model is optimized in regards to the pocket volume, the restraints will only minimally change the pocket geometry, but a broader bandwidth of different pocket volumes will be observed (see Appendix A).

In RosettaGPCR, the distribution of different orthosteric binding site volumes is narrow. Notably, when ranking the models according to their total energy, the models with similarly dense pockets are ranked best. With the applied RosettaGPCRPocketSize algorithm, the volume distribution is significantly broader, and a volume ensemble is generated. A variety of different pocket sizes is sampled including pockets with significantly smaller and larger volumes (Figure 3B).

For every model generated with RosettaGPCR and RosettaGPCRPocketSize methods, the RMSD value relative to the experimental reference GPCR structure was calculated as well. We note, however, that the all-structure RMSD value as a metric of model accuracy is of secondary importance in this modeling task, because RosettaGPCRPocketSize is intended to build a biophysically realistic ensemble of pocket geometries. An RMSD value in a similar range as for RosettaGPCR was considered acceptable. Indeed, compared to RosettaGPCR, only a small increase in the RMSD values was observed (Figure 3D), which was around 0.5 Å for most cases and not larger than 1 Å. For AA1AR and CXCR2, the model accuracy even increased. To focus on the pocket area, the RMSD value was also calculated for the upper TM bundle region, using the same ten residues that we used for the volume calculation. Again, a small RMSD deterioration was observed but no drastic change (see Appendix A).

Furthermore, we tested whether the selected restraints could correct the pocket shrinkage observed in the original RosettaGPCR models. Therefore, the final models from RosettaGPCR were minimized with and without restraints from the RosettaGPCRPocketSize protocol. Relaxation without restraints resulted in even more shrinkage of the volume in two-thirds of the cases up to more than 200 Å^3^, but sometimes the volume also increased. With RosettaGPCRPocketSize restraints, the shrinkage was reversed, and the binding site expanded by up to 340 Å^3^. This shows a possible application of RosettaGPCRPocketSize for correcting existing homolog models or for simulating the flexibility of the binding pocket for a determined receptor structure. The results can be seen in Appendix A.

### 2.4. Comparison to Other GPCR Models and Databases

In the next step, we compared models generated with RosettaGPCRPocketSize to models of the same receptor from other GPCR modeling databases and AlphaFold2. To avoid biases by including receptors that were used to derive our customized restraints, we only used GPCR structures deposited after downloading the previous dataset: growth hormone secretagogue receptor (GHSR, PDB: 6KO5) [73], neuropeptide Y2 receptor (Y2R, PDB: 7DDZ) [74], oxytocin receptor (OTR, PDB: 6TPK) [75], and gonadotropin-releasing hormone receptor (GNRH, PDB: 7BR3) [76]. For RosettaGPCRPocketSize, the five best-scoring models of each receptor after filtering were evaluated. For comparison, we also generated models with RosettaGPCR without restraints, including an additional relaxation step after template hybridization, and selected the best five models by total Rosetta score as detailed before (Figure 4A).

With RosettaGPCR, the volume difference to the determined structure was −126 to −699 Å^3^ for the four GPCRs tested. With RosettaGPCRPocketSize the shrinkage was reduced: for GNRH the volume change was −182 Å^3^ on average, and for GHSR and Y2R the volume was slightly increased, but still below 200 Å^3^. Only for OTR was a pocket volume reduction of −271 Å^3^ observed, but the distribution of generated pocket volumes was generally larger. When no restraints were applied in RosettaGPCRPocketSize (RosettaGPCR with an additional relaxation step), an even bigger volume reduction was observed, highlighting the effect of the relaxation step, which can have a negative impact on the pocket volume as demonstrated in Figure 2C. For individual models, the shrinkage ranged from −125 to −743 Å^3^. For the selected best five models on average, the shrinkage was −244 Å^3^ for GHSR and up to −612 Å^3^ for GNRH. In the database models, a more or less pronounced shrinkage of the orthosteric binding pocket was visible. This effect was seen across all databases and receptors, no single database was superior to all others in terms of recovering the experimental structure’s pocket size.

For GHSR, all database models were in the acceptable range of ±200 Å^3^, whereas RosettaGPCRPocketSize increased the volume, comparable to the model from GPCR-SSFE. The AlphaFold2 model has a slightly increased volume of +52 Å^3^. All database models of GNRH had a volume decrease of more than −380 Å^3^, and only with AlphaFold2 and with RosettaGPCRPocketSize could a volume difference in the tolerable range of ±200 Å^3^ be achieved. Thus, our new protocol was significantly superior to other homology modeling methods for this example. For OTR, on the other hand, a clear shrinkage of around −271 Å^3^ on average was observed. We note that OTR has an enlarged binding pocket compared to other non-peptide antagonist-bound neuropeptide GPCRs [75], which was a fact that made modeling more difficult. Models from GPCR-TASSER and GPCRdb, however, had pocket volumes similar to the native structure, but AlphaFold2 failed to capture a realistic pocket volume for OTR. The last receptor investigated, Y2R, showed a variety of different pocket volumes in the different models. Models from RosettaGPCR and GPCR-SSFE showed a significant shrinkage, whereas models from GPCR-I-TASSER, AlphaFold2, and GPCRdb had a small volume increase or decrease. RosettaGPCRPocketSize increased the pocket volume in the realistic range, too. Overall, AlphaFold2, GPCR-Tasser, GPCRdb, and RosettaGPCRPocketSize had three out of four receptor models in the desired pocket range. When summing up the absolute volume differences to the experimental reference over all models of the four GPCRs investigated, AlphaFold2 had the best performance (483 Å^3^ difference), followed by RosettaGPCRPocketSize (722 Å^3^ difference) and, lastly, GPCR-SSFE (1694 Å^3^ difference).

When looking at the all-structure RMSD, AlphaFold2 dominates the benchmark set with the lowest values overall. RosettaGPCR with the additional relaxation step produced worse results than the updated algorithm RosettaGPCRPocketSize. The models of GNRH and OTR in GPCR-SSFE were without loops and, therefore, a comprehensive comparison was not possible (Figure 4B).

### 2.5. Improvement for Ligand Docking in Drug Discovery

A high-resolution model of a GPCR including a high-quality representation of its respective orthosteric binding site is essential for adequate structure-based drug-discovery studies. An artificial, collapsed binding site hinders virtual docking experiments and biases the results towards smaller molecules while discriminating larger ones due to the pocket collapse. To examine the results of RosettaGPCRPocketSize, we designed a docking benchmark in which known ligands with a larger molecular weight of 400–700 Da, compared to common drug-likeness criteria (MW < 500 Da), were used. Compounds were taken from screening datasets deposited in PubChem [77]. Ligands that were confirmed actives for a receptor were gathered, their molecular weight calculated, and five ligands within the desired MW range were randomly selected. The ligands were docked to the models created with RosettaGPCR and RosettaGPCRPocketSize or were downloaded and prepared from the respective databases, and the number of successfully docked poses was compared. A successful or productive docking pose was defined as a pose with a Rosetta interface score below -6 REU. Unfortunately, as the calculated binding energy is dependent on the selected receptor and ligand size, there is no general applicable cutoff for active ligands. Our goal of the success metric was to compare the modeling of productive binding pocket geometry for ligand docking, the generation of low-energy docking poses, and the avoidance of steric clashes due to a collapsed binding pocket.

Without applying volume restraints in RosettaGPCR modeling, the ligands rarely fit into the binding site of any relaxed RosettaGPCR model in the subsequent ligand docking calculation. Overall, 20 receptor–ligand pairs were investigated, and less than 20% resulted in successful docking results. Thus, many non-fitting but experimental active ligands would be discarded in virtual screening experiments. The results obtained for the RosettaGPCR model prepared without relaxation were similar in the case of GNRH. The pocket collapse in GNRH was the largest, and the resulting number of docked poses was the lowest. This shows the importance of the pocket size for docking endeavors. The pocket volume shrinkage can introduce a bias towards discriminating larger ligands (see Figure 5B). However, this dependence is not always the case. While for Y2R, a clear trend between small pocket volumes and the low number of docking poses is observed, no such dependence is seen for OTR. For this GPCR, the model from GPCRdb had the highest pocket volume, but this is not reflected in the docking numbers. Interestingly, in OTR the number of successfully docked ligands is comparable for RosettaGPCR and RosettaGPCRPocketSize while the relaxed structures without restrains yielded almost no successful docking poses. This highlights the effect of the additional relaxation step in collapsing the binding pocket. The number of successfully docked poses for GHSR and Y2R is significantly higher in RosettaGPCRPocketSize. The number of successfully docked ligand poses for database models varies across receptors and databases. Often, they are comparable to the results obtained with RosettaGPCRPocketSize, sometimes in a lower or even higher range.

For RosettaGPCRPocketSize models, the success rate ranged from 5% to over 90%. This reflects the variability of pocket sizes for these models. Here, the number of successfully docked poses is comparable to the number in the determined structure. In GNRH, the experimental reference structure yields less successful docking poses than in the case of RosettaGPCRPocketSize or database models even though it was determined with a ligand with 631.6 Da weight [76]. However, the orthosteric binding site might discriminate further larger ligands and different pocket representations are needed to overcome this bias.

It must be highlighted that some database models had missing loops, which represents an advantage for docking as the ligand can move and dock in a more unhindered manner. As a result, the number of successfully docked poses is artificially increased. Therefore, the results for the GNRH and OTR model of GPCR-SSFE should be treated with caution. Especially in OTR, the GPCR-SSFE model showed the largest pocket shrinkage, but the docking is not hindered due to the open access without loops.

All in all, RosettaGPCRPocketSize models are on par with database models. Compared to the original protocol, the search space for ligand docking is improved. When no restraints are implemented, larger compounds are highly disfavored. However, as a side note, it must be mentioned that the ligands were selected without regard to binding position. As GPCRs possess several binding sites, including allosteric sites deep in the binding pocket or at the membrane site, the ligand could interact not in the selected pocket [78]. Often, the exact binding area is not known or sparsely investigated. However, for the goal of this benchmark, the binding site was presumed to be in the orthosteric binding site, and the general docking results were compared. Additionally, while enrichment factor or ROC are favorable metrics for successful virtual screening methods, we focused in this benchmark on the generation of productive binding poses without steric clashes. The distinguishing of binders from non-binders is a non-trivial task that is not reached in any docking method. As Rosetta excels in predicting near-native binding poses for ligands, we focused on benchmarking the improvement of binding pocket generations with RosettaGPCRPocketSize.

## 3. Discussion

Proteins and their ligands are highly flexible, and a single determined structure or a constructed model is only a snapshot of their possible conformations. An equilibrium of different conformational ensembles over time is always present and can be affected by a plethora of various binding partners. As GPCRs are a group of conformationally highly flexible transmembrane proteins, the assumption that receptors have one single orthosteric binding pocket conformation is not adequate. Accurate modeling of the orthosteric binding site for GPCRs is of the highest interest in drug discovery endeavors. Applying a general protocol to generate a static pocket disregards the structural perspective of the ligand binding and the molecular basis of activation. Considering the flexibility of the extracellular loops and of the pocket itself, modeling the binding pocket in multiple probable conformations will more likely capture the natural flexibility of GPCRs and their ligand interactions. We designed a homology modeling protocol in Rosetta, termed RosettaGPCRPocketSize, which uses a curated restraint set to achieve the modeling of GPCRs with flexible binding pockets as a starting point for drug discovery experiments.

While the structures of apo state receptors tend to have a larger pocket volume, upon ligand binding, the pocket shrinks to enforce tighter interactions between ligand and protein. We observed a shrinkage from apo to ligand-bound of the inactive state for ADRB1, EDRB, and ACM4 of 288–361 Å^3^, equal to a reduction of 11–12% of the apo state volume. Furthermore, in the active state, the pocket volume shrinks even more. Here, a shrinkage of again 92–496 Å^3^ can be observed, equal to a reduction of 4–17% of the inactive volume. However, there is not one single value for the pocket volume of various experimentally determined structures of the same state, but the volume varies by around ±100 to 150 Å^3^. To model this bandwidth, a tetrahedral restraint set was introduced.

Without the restraint set applied in RosettaGPCR, often a pocket shrinkage is observed, up to ~600 Å^3^, and ligand docking is hindered. With our designed restraints and modeling protocol, the shrinkage was reduced, and the volume range increased. Furthermore, even structures showing a shrunken pocket can be corrected by relaxing the structure with restraints from RosettaGPCRPocketSize. Therefore, this protocol set can be combined with already prepared RosettaGPCR models in order to improve the resulting pockets and to generate other possible binding pockets with different volumes. Compared to other GPCR models from GPCRdb, GPCR-I-Tasser, and GPCR-SSFE, our method improved the pocket volume significantly. Only AlphaFold2 was able to produce high-quality pockets, as well.

Different approaches for ligand binding site refinement have been published. In these pipelines, an apo structure is refined by molecular dynamics (MD) simulation toward a reliable holo protein structure, usable for ligand docking. Knowledge-based restraints are applied in the short simulation to define and open the binding site [42,43]. An MD simulation is time and computational power expensive, and this approach has only been shown for apo-to-holo models, not for a refinement of already prepared models of GPCRs. Our method is able to produce models for docking experiments more rapidly and reliably. It has to be mentioned that the new modeling pipeline and algorithm is of a computational nature and includes, currently, no further experimental validation. While computational modeling and simulation offer many benefits, a determined structure represents an excellent starting point for drug discovery. The here-presented pipeline can be combined with the experimental structure to enhance the sampling space but built models need to be validated against the starting structure and its given binding pocket geometry.

In recent times, the research on biased signaling in GPCRs has gained more and more interest. By regulating the biological signaling and the corresponding pathways in a precise way, new advanced drugs with superior efficiency and/or reduced adverse effects can be explored. Moreover, with more and more available experimental structures, the understanding of how affinity and signaling pathways are determined is improving [61,79]. The here-presented dataset focused on a general approach for inactive class A GPCRs. However, it is possible to adopt the restraint set for a specific class of receptors, ligands, and estimated binding pocket size. With RosettaGPCRPocketSize, we provide an innovative way to tackle the challenge of GPCR orthosteric binding site modeling and analysis.

## 4. Materials and Methods

### 4.1. Original RosettaCM Protocol and Improved Scripts

In this study, we used the optimized RosettaCM [32] protocol for GPCRs, RosettaGPCR [39], with the provided input files for template selection, alignment, and threading scripts. The final script can be found in Protocol Capture S1. In addition to the input files provided with the RosettaGPCR paper, receptor-specific restraint files were built based on the here-presented algorithm. An additional relaxation (fast relax) step after the RosettaCM Hybridize Mover was added to the protocol. Furthermore, the steps for adding and clearing restraints as well as reading the restraint weights were modified.

### 4.2. Definition of the Binding Pocket and Pocket Volume Calculation

The orthosteric binding pocket is located between the seven transmembrane helical bundles and opens toward the extracellular space. The most important helices for forming the pocket boundaries are TMs 2, 3, 5, 6, and 7. Identifying the binding pocket and calculating its volume in an automatic way is a non-trivial and complex task. Several computational methods try to solve this three-dimensional problem including RosettaHoles with cavity balls and Delaunay triangulation [80,81]. In this study, we chose to use Delaunay triangulation because of its simplicity and calculation speed. The topology is divided into simplices (triangles in two dimensions, tetrahedrons in three dimensions) and calculates the respective surface or volume for each simplex and the sum for the whole geometrical construct [82,83]. With the help of triangulation, the volume enclosed between specific points in space, in this case, the Cartesian coordinates of selected residues in the GPCR, can be calculated. To generalize the volume calculation independent of amino acid types in the different receptors and develop a generalized protocol for restraining the pocket volume in Rosetta, we used the α-carbons (C_α_) of the backbone to calculate the pocket volume. Furthermore, using the C_α_ positions ensured that the same restraints can be used in Rosetta’s centroid mode, in which only the backbone atoms are present and the sidechain is modeled as a single bead, as well as in Rosetta’s full atom mode. Ten surrogate residues located on TMs 2, 3, 5, 6 and 7 were selected from the set of thirty-four conserved pocket surrounding residues.. These residues are located at one of the highest and lowest positions on one of the five TM-helices and frequently interact with a ligand atom in determined structures, therefore defining the limit of the pocket.

### 4.3. Definition of Volume Restraints and Restraint Parameter Selection

In Rosetta, a restraint definition includes the atoms defining the distance (or angle or dihedral) being measured, a reference value, to which the measured distance (or angle or dihedral) is compared, and a mathematical function to translate the deviation between measured and reference value into a score. In addition to single restraint definitions, nested restraints, which group multiple single restraints, are possible. These can be combined with different potential functions that calculate the score of a restraint. In the following example a distance definition is shown:AtomPair CA X CA Y SCALARWEIGHTEDFUNC A SUMFUNC 2 HARMONIC B C CONSTANTFUNC D

The restraint definition contains the selected residue numbers X and Y according to Rosetta’s residue numbering scheme (starting at 1 for the first residue in structural model), the reference distance *C* between the selected AtomPair, a weight *A* defined by the SCALARWEIGHTEDFUNC keyword, the steepness of the chosen harmonic potential as *B*, and a constant score offset *D* following the keyword CONSTANTFUNC. Summarized, the mathematical function of the restraint can be written as:(1)fx=A×((measured distance−C B)2+D)

Furthermore, different functional forms of the potential can be selected, including linear, harmonic, flat harmonic, and sigmoid functions. In the benchmark set, the standard harmonic function combined with a constant function offset was used because it performed best in reducing pocket shrinkage in the structural modeling. The standard harmonic function performed best among all functions investigated, including flat harmonic function and a combination of two sigmoid functions.

As GPCR structures are dynamic in the area of the orthosteric binding site as evidenced by our analysis of the pocket volumes of multiple GPCR structures, a fixed restraint would not be optimal. To allow some variability of the created GPCR homology models in terms of their pocket sizes while maintaining physically realistic binding pocket geometries, we systematically varied all parameters of the restraint definitions to find the ideal value range. The following parameter ranges were found to be optimal for modeling a wide range of different pocket geometries and volume ensembles that fulfilled the naturally observed variances of GPCR pockets:$$\begin{array}{l} \mathrm{SCALARWEIGHTEDFUNC}:\;2–4\\ \mathrm{Distance C}:\;1.03–1.07\;\mathrm{times the determined distance in}\;Å\\ \mathrm{HARMONIC}:\;0.4–0.8\\ \mathrm{CONSTANTFUNC}:\;-1\;\mathrm{to}\;-10 \end{array}$$


An example of one tetrahedron restraint set for ADRB1 is given in the following:
$$\begin{array}{l} \mathrm{AtomPair CA}66\;\mathrm{CA}\;98\;\mathrm{SCALARWEIGHTEDFUNC}\;3\;\mathrm{SUMFUNC}2\\ \mathrm{HARMONIC}\;23.7864\;0.7\;\mathrm{CONSTANTFUNC}-8\\ \mathrm{AtomPair CA}66\;\mathrm{CA}\;176\;\mathrm{SCALARWEIGHTEDFUNC}\;3\;\mathrm{SUMFUNC}2\\ \mathrm{HARMONIC}\;25.6431\;0.7\;\mathrm{CONSTANTFUNC}-8\\ \mathrm{AtomPair CA}66\;\mathrm{CA}\;308\;\mathrm{SCALARWEIGHTEDFUNC}\;3\;\mathrm{SUMFUNC}2\\ \mathrm{HARMONIC}\;19.5077\;0.7\;\mathrm{CONSTANTFUNC}-8\\ \mathrm{AtomPair CA}98\;\mathrm{CA}\;176\;\mathrm{SCALARWEIGHTEDFUNC}\;3\;\mathrm{SUMFUNC}2\\ \mathrm{HARMONIC}\;19.0209\;0.7\;\mathrm{CONSTANTFUNC}-8 \end{array}$$
$$\begin{array}{l} \mathrm{AtomPair CA}98\;\mathrm{CA}\;308\;\mathrm{SCALARWEIGHTEDFUNC}\;3\;\mathrm{SUMFUNC}2\\ \mathrm{HARMONIC}\;26.8934\;0.7\;\mathrm{CONSTANTFUNC}-8\\ \mathrm{AtomPair CA}176\;\mathrm{CA}\;308\;\mathrm{SCALARWEIGHTEDFUNC}\;3\;\mathrm{SUMFUNC}2\\ \mathrm{HARMONIC}\;22.3116\;0.7\;\mathrm{CONSTANTFUNC}-8 \end{array}$$


### 4.4. Comparative Modeling and Modeling Pipeline

For each GPCR, 100 models were created with RosettaCM through Rosetta XML scripts [84]. The modeling protocol of RosettaGPCRPocketSize can be split into four steps: First, the already prepared templates and sequence alignments of RosettaGPCR are selected, restraints are generated for each receptor individually, and the parameters for each tetrahedron set are selected. In the next step, the Hybridize Mover of RosettaCM with a membrane scoring function and the two tetrahedral restraints equaling 2 × 6 distance restraints are utilized. A following relaxation step, with the tetrahedron restraints energy, improves the model structure. Finally, the models are rescored without restraints and the volume is calculated. For each target receptor, the median pocket volume is calculated from the volume distribution, and the models falling within a ±200 Å^3^ range to the median are selected for further analysis. The five best-scoring models were used for calculating the RMSD and pocket volume.

### 4.5. Compilation of the Benchmark Set

All experimentally determined class A GPCR structures available as of November 2020 were downloaded from GPCRdb [46,85] and cleaned. The coordinates corresponding to fusion proteins, ligands, and other non-amino acid residues were deleted from the PDB files. For each selected receptor, if available, the inactive structure with the best experimental resolution was selected. Furthermore, the orthosteric binding pocket volumes of the determined structure and the model from the RosettaGPCR database (http://www.meilerlab.org/index.php/gpcrmodeldb (accessed on 16 July 2020)) were calculated and the volume difference investigated. The benchmark set included receptors from various class A GPCR subfamilies with differently large pocket volumes. Furthermore, the observed pocket shrinkage covers a broad range. This benchmark includes receptors with their respective determined structure included as template in RosettaGPCR and some without. Another benchmark set focused solely on newly determined structures, which were not present in the RosettaGPCR database. The results were compared to models produced without the restraints taken from RosettaGPCR, GPCRdb (https://gpcrdb.org/structure/homology_models (accessed on 25 February 2021)) [85], GPCR-I-TASSER (https://zhanglab.ccmb.med.umich.edu/GPCR-I-TASSER/ (accessed on 25 February 2021)) [86], GPCR-SSFE 2.0 (http://www.ssfa-7tmr.de/ssfe2/ (accessed on 25 February 2021)) [87], and AlphaFold2 (https://alphafold.ebi.ac.uk/ (accessed on 14 August 2022)) [31,88].

### 4.6. Ligand Docking

For ligand docking, the RosettaLigand [89,90] protocol was utilized. The models were downloaded from the mentioned databases, and the five lowest energy models for each receptor from RosettaGPCRPocketSize with and without restraints, and the experimentally determined structures after relaxation in Rosetta were selected for ligand docking experiments. From the PubChem database (https://pubchem.ncbi.nlm.nih.gov/ (accessed on 26 February 2021)) [77] all identified hits for the respective receptor were downloaded and filtered. From all identified ligands with a molecular weight of 400–700 Da, five compounds were randomly selected, and conformational ensembles were computed with the BioChemicalLibrary (BCL) [91,92,93]. A total of 1000 docking simulations for each structure were performed and the number of successfully docked poses was counted and compared. An interface_delta_X score cutoff of −6 REU or lower was used as a criterion to define a successful case.

## Figures and Tables

**Figure 1 ijms-24-07788-f001:**
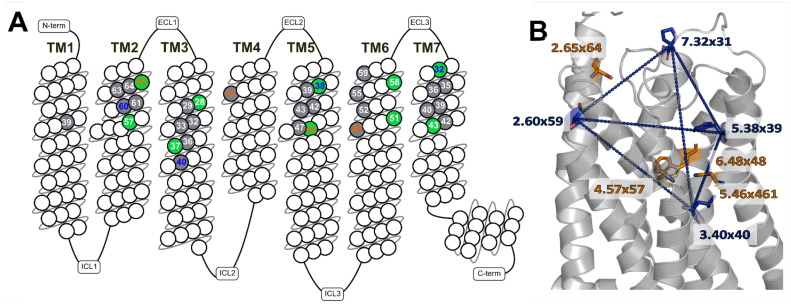
Selected residues defining the orthosteric binding pocket in bovine rhodopsin (OPSD, PDB: 1U19) and used for building tetrahedron restraints: (**A**) All selected 34 pocket residues are highlighted in a snakeplot of rhodopsin. The selected residues point inwards to the orthosteric binding site and are accessible to the ligand. These residues define the orthosteric binding pocket and its volume with their backbone and sidechains. A subset of ten residues (green filled) is used as surrogate coordinates for triangulation to calculate the orthosteric binding pocket volume. Residue numbers are according to BW numbering. Two tetrahedrons are selected in blue and orange numbers. (**B**) Two groups of four selected residues each define two tetrahedrons (highlighted in the rhodopsin structure). One tetrahedron with its distances is visualized.

**Figure 2 ijms-24-07788-f002:**
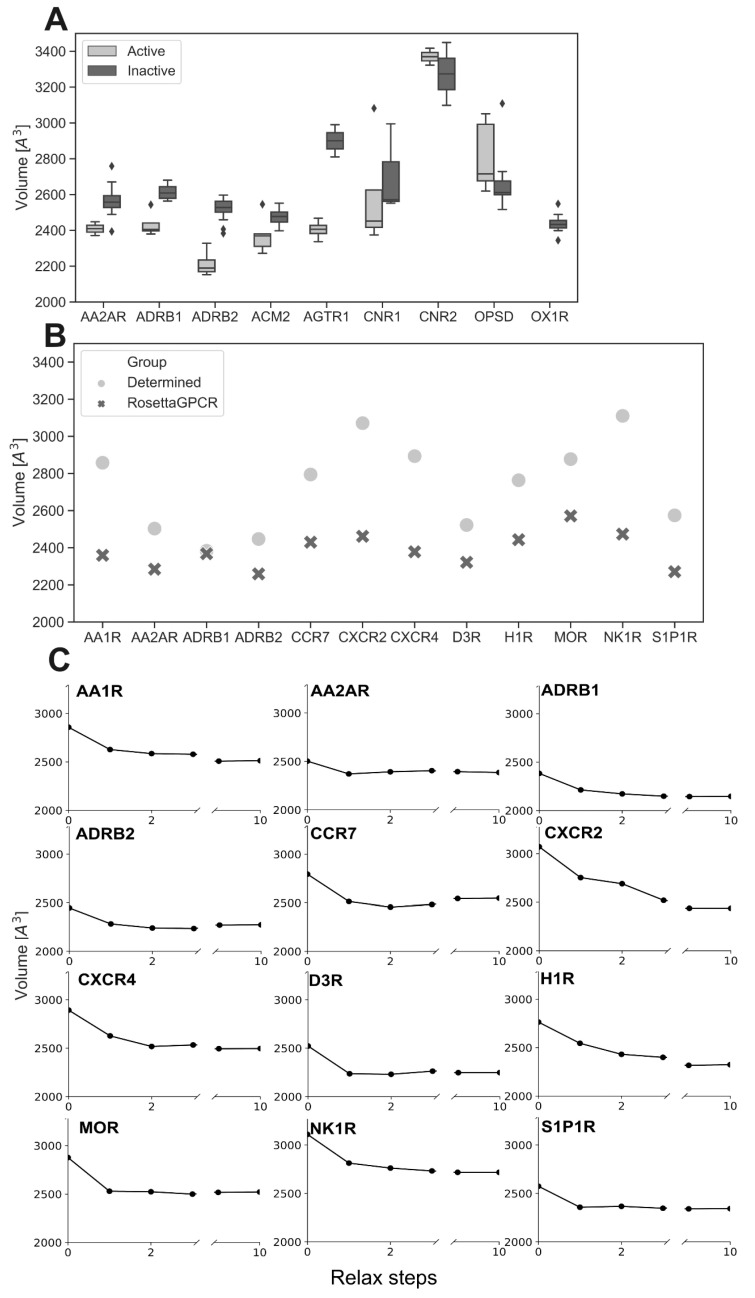
Distribution of orthosteric binding pocket volumes of GPCRs and effect of pocket shrinkage in RosettaGPCR and structure relaxation with Rosetta: (**A**) The pocket volumes of GPCRs with more than one inactive and active structure were compared. Pocket size distributions have a variance of nearly ±200 Å^3^ for the investigated receptors. Diamonds highlighting outlier data points. (**B**) Pocket volumes of class A GPCR structures from different subfamilies and of corresponding RosettaGPCR models. In the case of multiple experimental structures, the one with the best resolution was used for comparison. The smallest volume change is found for ADRB1 (16 Å^3^) and the largest change is seen for NK1R (637 Å^3^). (**C**) Experimental structures from B were repetitively energy-minimized with the Rosetta relax protocol. Circles representing the calculated volume, while the connecting lines emphases the decrease. The pocket volume rapidly decreases and stays constant after three steps of relaxation.

**Figure 3 ijms-24-07788-f003:**
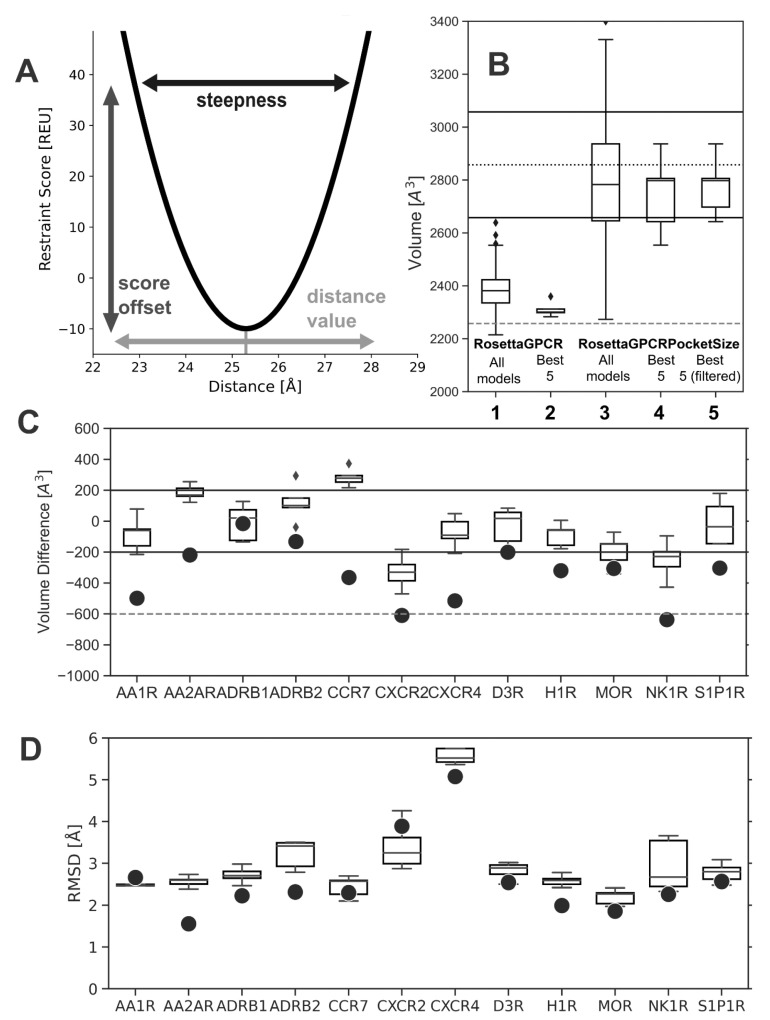
Volume differences in GPCR models generated with RosettaGPCR and RosettaGPCRPocketSize with respect to their experimental structures and effect of the improved algorithm with restraints: (**A**) Functional form of the distance restraint potential. The parameters of the potential (steepness, offset, distance center) are selected from an optimized parameter range at the beginning of each run. (**B**) Pocket volumes of homology models generated for AA1R: (1) RosettaGPCR—100 models, (2) RosettaGPCR—5 top models, (3) RosettaGPCRPocketSize—100 models, (4) RosettaGPCRPocketSize—5 top models, (5) RosettaGPCRPocketSize + Volume filter—5 top models. (**C**) Pocket volume differences relative to the experimental reference structure for the best model generated with RosettaGPCR (single circle) and the best five models generated with RosettaGPCRPocketSize (box plot) for the GPCRs listed. Diamonds are demonstrating outlier data points. A volume difference below −200 Å^3^ indicates a pronounced pocket shrinkage, whereas a value above +200 Å^3^ marks an enlarged pocket volume. Cases with a volume difference smaller or equal to ±200 Å^3^ fall in the range of the natural pocket size variation, which is observed for known GPCR structures. Three times this difference is considered a failed pocket construction. (**D**) RMSD relative to the experimental reference structure for the best model generated with RosettaGPCR (single circle) and the best five models generated with RosettaGPCRPocketSize (box plot) for the GPCRs listed. Only a minor increase in the RMSD with RosettaGPCRPocketSize compared to RosettaGPCR is observed.

**Figure 4 ijms-24-07788-f004:**
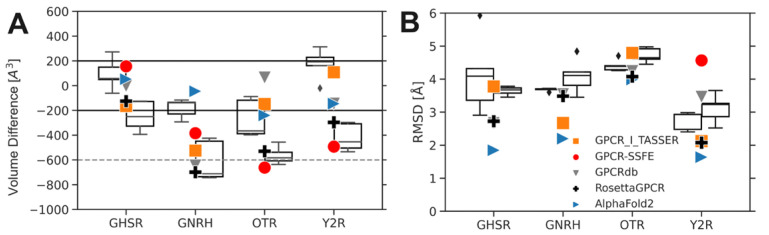
Comparison of the pocket volume and RMSD of GPCR models produced with RosettaGPCRPocketSize with those from different GPCR model databases. The structures of four GPCRs, which were not contained in the previous benchmark set, were selected and their respective models from other GPCR databases were downloaded. RosettaGPCRPocketSize with and without restraints was applied and the best five models from 100 generated structures were selected according to the algorithm: (**A**) Volume difference to the determined structure. A volume difference below −200 Å^3^ indicates a pronounced pocket shrinkage, whereas a value above +200 Å^3^ marks an enlarged pocket volume. RMSD. Left boxplot—RosettaGPCRPocketSize, right boxplot—RosettaGPCR with an additional relaxation step, blue triangle right—AlphaFold2, orange square—GPCR-Tasser, red circle—GPCR-SSFE, grey triangle down—GPCRdb, black plus—RosettaGPCR (**B**) RMSD values to the determined structure. Note that the AlphaFold2-predicted model has the lowest RMSD. Left boxplot—RosettaGPCRPocketSize, right boxplot—RosettaGPCR with an additional relaxation step, blue triangle right—AlphaFold2, orange square—GPCR-Tasser, red circle—GPCR-SSFE, grey triangle down—GPCRdb, black plus—RosettaGPCR. Diamonds are demonstrating outlier data points.

**Figure 5 ijms-24-07788-f005:**
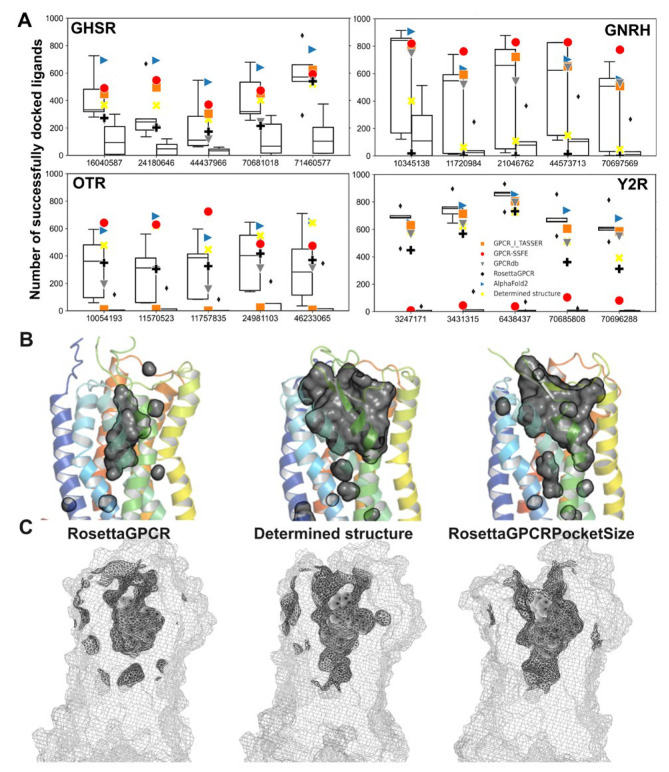
Number of successful ligand docking cases for GPCR homology models generated with either RosettaGPCR or RosettaGPCRPocketSize and from other GPCR databases for four different receptors and the change in pocket volume inY2R: (**A**) Ligands (labeled by their PubChem ID) were docked 5 × 200 times in each model. For RosettaGPCR and RosettaGPCRPocketSize the best five models according to the described method were taken. Left boxplot—RosettaGPCRPocketSize, right boxplot—RosettaGPCR with an additional relaxation step, yellow cross—determined structure, blue triangle right—AlphaFold2, orange square—GPCR-Tasser, red circle—GPCR-SSFE, grey triangle down—GPCRdb, black plus—RosettaGPCR. (**B**) The structure of Y2R without ligand clearly indicates the difference in pocket volume and size for RosettaGPCR and RosettaGPCRPocketSize in comparison to the determined structure. Inverse surface plot in PyMOL highlights the shrunken pocket in RosettaGPCR models. (**C**) When a ligand is docked to the receptor structures with RosettaLigand, side chain movement opens the pocket slightly, but as seen in the inverse negative mesh representation, the binding site does not fit the ligand (in spheres representation). Diamonds are demonstrating outlier data points.

**Table 1 ijms-24-07788-t001:** Volume of various receptors in inactive, ligand-free, or inactive, ligand-bound conformations. The volume of single structures of ACM4, ADRB1, EDNRB, and OPSD with and without ligands was calculated and compared. A pocket-size shrinkage was observed in the presence of a ligand.

Receptor	PDB Number	Volume (Å^3^)	Conformational/Ligand State	Volume Difference (Å^3^)
ACM4	6KP6	2925	Inactive/Ligand-free (Apo)	361
ACM4	5DSG	2564	Inactive/Ligand-bound	
ADRB1	4GPO	2694	Inactive/Ligand-free (Apo)	287
ADRB1	7JJO	2407	Inactive/Ligand-bound	
EDNRB	5XPR	2716	Inactive/Ligand-free (Apo)	305
EDNRB	6IGK	2411	Inactive/Ligand-bound	
OPSD	1U19	2611	Inactive/Ligand-free (Apo)	−16
OPSD	2I37	2627	Inactive/Ligand-bound	

**Table 2 ijms-24-07788-t002:** Selected C_α_ residues and corresponding distances. Two sets of four residues each were selected to define two tetrahedrons in the binding pocket. Each tetrahedron restraint is defined by six edges with distances listed in the right column. The distances are the median distance of all investigated receptors.

Restraint	Tetrahedron	Residue 1	Residue 2	Distance (Å)
1	1	2.60 × 59	3.40 × 40	22.50
2		2.60 × 59	5.38 × 39	24.25
3		2.60 × 59	7.32 × 31	18.78
4		3.40 × 40	5.38 × 39	17.94
5		3.40 × 40	7.32 × 31	25.35
6		5.38 × 39	7.32 × 31	20.30
1	2	2.65 × 64	4.57 × 57	23.08
2		2.65 × 64	5.46 × 461	25.35
3		2.65 × 64	6.48 × 48	19.92
4		4.57 × 57	5.46 × 461	10.76
5		4.57 × 57	6.48 × 48	17.34
6		5.46 × 461	6.48 × 48	11.23

## Data Availability

Data are contained within the article or Appendix A. Furthermore, all scripts and the protocol capture are available at: https://github.com/FabianLiessmann/RosettaGPCRPocketSize (accessed on 27 March 2023).

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
