# Peer review of "Improving the Modeling of Extracellular Ligand Binding Pockets in RosettaGPCR for Conformational Selection"

_ijms, 2023, doi:10.3390/ijms24097788_

Round 1

Reviewer 1 Report

Liessmann and coworkers report RosettaGPCRPocketSize, a homology modeling tool dedicated to drug discovery purposes. The manuscript is generally well-written and easy to read. The work reported in the manuscript seems relevant, and although the resulting predictions are on average not better than the AlphaFold models, it still has the potential as an alternative tool to AlphaFold for pocket ensemble generation. However, the manuscript has a critical issue that makes me hard to recommend it in its current form.

The major issue I would like to point out is how the authors presented their protein model accuracy for ligand docking/screening (section 2.5 & Fig 5). The authors judged "successful docking poses" based on calculated binding energy, but what is the basis? Is there any study showing -6 REU can be a generally applicable criterion for binder/non-binder classification? Is there any reason why the authors did not report direct evidence of docking success by ligand RMSD? If the purpose was to show its screening power instead of structural analysis, then the authors should have at least included virtual screening results (not just binder energy values) by adding ROC or enrichment factor analysis. 

Minor point: Several references have typos or missing information -- ref 43, 90, 91. Please fill in.

Author Response

The major issue I would like to point out is how the authors presented their protein model accuracy for ligand docking/screening (section 2.5 & Fig 5). The authors judged "successful docking poses" based on calculated binding energy, but what is the basis? Is there any study showing -6 REU can be a generally applicable criterion for binder/non-binder classification? Is there any reason why the authors did not report direct evidence of docking success by ligand RMSD? If the purpose was to show its screening power instead of structural analysis, then the authors should have at least included virtual screening results (not just binder energy values) by adding ROC or enrichment factor analysis. 

We agree with the reviewer that ligand RMSD and enrichment factor are preferred metrics for evaluating the predictive power of a virtual screening method. Overall, however, the goal of this section of the results was to showcase that the improved pocket geometry of our Rosetta models leads to docking poses with more productive protein-ligand interactions ligands with larger size between 400 and 800 Da. This is indicated by more negative docking scores compared to the docking results obtained with the default RosettaGPCR models. Evaluation by ligand RMSD was, however, not possible because there were no X-ray structures of ligands from this MW category for the set of GPCRs studied.

Furthermore, we agree that enrichment factor or ROC would show the predictive screening power of ligand docking performed on RosettaGPCRPocketSize models. However, it is known that Rosetta ligand docking, as well as other docking algorithms, are not accurate in distinguishing actives from non-actives despite being accurate in predicting a near-native binding pose for actives. The classification for binders and non-binders is, as far as we know, a general difficulty and is not completely achieved by any docking methods. Thus, we did not attempt to tackle this additional challenge in the current study. With the so-called success metric, we aimed to compare the extent of productive binding pocket geometries for ligand docking, low energy docking poses, and the avoidance of steric clashes due to collapsed binding pockets. Therefore, we chose a negative binding score to label good poses and defined them as “successful” docking poses, meaning a productive docking result. The score cutoff of -6 REU was chosen because this value is an indicator that the ligand fits well in the binding pocket and is free of steric clashes.

We thank the reviewer for this helpful comments, which we have addressed now in the discussion in section 2.5:

“A high-resolution model of a GPCR including a high-quality representation of its respective orthosteric binding site is essential for adequate structure-based drug-discovery studies. An artificial, collapsed binding site hinders virtual docking experiments and biases the results towards smaller molecules while discriminating larger ones due to the pocket collapse. To examine the results of RosettaGPCRPocketSize, we designed a docking benchmark in which known ligands with a larger molecular weight of 400-700 Da, compared to common drug-likeness criteria (MW < 500 Da), were used. Compounds were taken from screening datasets deposited in PubChem [77]. Ligands that were confirmed actives for a receptor were gathered, their molecular weight calculated, and five ligands within the desired MW range were randomly selected. The ligands were docked to the models created with RosettaGPCR and RosettaGPCRPocketSize or that were downloaded and prepared from the respective databases, and the number of successfully docked poses was compared. A successful or productive docking pose was defined as a pose with a Rosetta interface score below - 6 REU. A score value of this size is usually a valid indicator that the ligand fits well in the binding pocket and has no steric clashes. Our goal of the success metric was to evaluate the extent of productive binding pocket geometries suitable for ligand docking, the generation of low energy docking poses, and the avoidance of steric clashes due to a collapsed binding pocket.

All in all, RosettaGPCRPocketSize models are on a par with database models. Compared to the original protocol the search space for ligand docking is improved. When no restraints are implemented, larger compounds are highly disfavored. However, as a side note it must be mentioned that the ligands were selected without regard to binding position. As GPCRs possess several binding sites, including allosteric sites deep in the binding pocket or at the membrane site, the ligand could interact not in the selected pocket [78]. Often, the exact binding area is not known or sparsely investigated. However, for the goal of this benchmark the binding site was presumed to be in the orthosteric binding site, and the general docking results compared. Additionally, while enrichment factor or ROC are favorable metrics for successful virtual screening methods, we focused in this benchmark on the generation of productive binding poses without steric clashes. The distinguishing of binders from non-binders is a non-trivial task that is not reached in any docking method. As Rosetta excels in predicting near-native binding poses for ligands we focused on benchmarking the improvement of binding pocket generations with RosettaGPCRPocketSize.”

Minor point: Several references have typos or missing information -- ref 43, 90, 91. Please fill in.

We thank the reviewer for pointing out the missing information in the three mentioned references. They were edited accordingly and updated in the current bibliography.

Reviewer 2 Report

A group of potential pocket geometries and sizes are created using the newly established approach, RosettaGPCRPocketSize, which is based on a database of experimental class A GPCR structures. By using helical residues that point inward and commonly interact with ligand molecules in the template structures, the authors establish the binding pocket. The class A GPCRs, which make up over 90% of all GPCRs and more than 80% of all determined structures, were the subject of particular attention. The number of GPCR homology models produced using either RosettaGPCR or RosettaGPCRPocketSize is compared to other GPCR databases for four distinct receptors and the modification of pocket volume in Y2R that successfully docked ligands. The work is generally well written, the techniques are well explained, and the results are thoroughly discussed. RosettaGPCRPocketSize can generate GPCR models with realistic orthosteric pocket volumes which will be a valuable tool for structure-based drug discovery applications. I recommend the paper the paper for publication after addressing my minor concern.

Since the study is fully computational, I request the author to make a statement of study limitation in the abstract/conclusion section of the manuscript.

Author Response

Since the study is fully computational, I request the author to make a statement of study limitation in the abstract/conclusion section of the manuscript.

The comments and summary of reviewer 2 are highly appreciated. We agree with the reviewer’s assessment and, as suggested, a short statement of the limitations of our study being of purely computational nature was included:

“It has to be mentioned that the new modeling pipeline and algorithm is of computational nature and includes currently, no further experimental validation. While computational modeling and simulation offer many benefits, a determined structure represents an excellent starting point for drug discovery. The here presented pipeline can be combined with the experimental structure to enhance the sampling space but built models need to be validated against the starting structure and its given binding pocket geometry.”

Round 2

Reviewer 1 Report

The revision addressed my concerns.